# Plant Species Richness and Invasional Meltdown in Different Parts of *Acer negundo* L. Secondary Range

**Denis I. Dubrovin** [1,*]**, Denis V. Veselkin** [1] **and Andrei P. Gusev** [2]

1    Institute of Plant and Animal Ecology of the Ural Branch of the Russian Academy of Sciences, Ekaterinburg 620144, Russia
2    Department of Geology and Geography, Francisk Skorina Gomel State University, 46019 Gomel, Belarus; official@mail.gsu.by
*    Correspondence: dubrovin_di@ipae.uran.ru

**Abstract:** To understand the alien plant invasion patterns, it is important to know if their consequences are equal in different regions, particularly in different parts of the secondary range. In this article, we estimated plant species richness in communities invaded by the North American tree *Acer negundo* L. in two remote regions: the Belarusian Polesia and the Middle Urals. We tested three hypotheses about: (1) decreased plant species richness in communities invaded by *A. negundo*; (2) presence of alien species in invaded communities—invasional meltdown hypothesis; and (3) different alien plant species richness in communities of different regions. In each region, 24 sample plots of 400 m$^2$ were described: 12 invaded and 12 non-invaded by *A. negundo*. The species richness of invaded plots decreased: total richness decreased by 21%–43%; the richness of herbaceous plants decreased by 24%–43%; and woody richness decreased by 8%–44%. The proportion of alien herbs in plots invaded by *A. negundo* increased by 35%. This is the first, although not exhaustive, confirmation of the invasional meltdown hypothesis for communities invaded by *A. negundo*. Alien herbs increasingly invaded communities of the Belarusian Polesia, alien trees—communities of the Middle Urals. Thus, regional geographical and floristic conditions should be considered when assessing the invasion consequences.

**Keywords:** plant diversity; ash-leaved maple; *Acer negundo*; plant invasions; invasional meltdown hypothesis; invasion levels





## 1. Introduction

One of the key consequences of alien (invasive) plant species distribution is a decrease in taxonomic species richness in recipient communities [1–5]. However, there are rare exceptions when plant invasions increase plant species richness [6]. Decreased species richness in invaded communities is related to the increased competitive ability of invasive species [7–10] or their ability to transform environmental conditions [11–16], including allelopathy [17,18].

Sometimes changes in environmental conditions caused by the invasion of one alien species may facilitate the invasion of other alien species. According to the invasional meltdown hypothesis (IMH) [10,19–21], the richness and proportion of alien species increase in invaded communities compared to non-invaded ones. The suggestions about the IMH mechanisms are associated with facilitation between alien species [19,20,22–24], i.e., it is suggested that alien dominants affect native plant species more than other alien species. The IMH does not have exhaustive empirical support. Sometimes facilitation effects may be weak [25], or instead, there may be increased competition, such as biotic resistance, which decreases the levels of invasion [19,26]. Another interpretation for the frequent coexistence of alien species suggests that such species are more common in native species-poor communities with high biomass and, as a result, with enough vacant niches [27].

Surely, the composition of the invaded communities depends not only on the direct or indirect effects of invasive dominance but also on the regional species pool richness and community completeness [28]. Actual species pools (community richness) are often positively linked to local (landscape) and regional species pools [29–32]. The richness of the regional species pool is generally related to the climatic features, area, and geographical heterogeneity of the region [33]. Community richness is related to propagule availability, community niche space size, and competition intensity [34] and depends on climatic, edaphic, and biotic factors as well as the biogeographic history of the region, landscape, and community [29,32]. For example, invasions in cold-climate regions were shown to have a greater impact on community richness, while invasions in warm-climate regions had a greater impact on landscape-level richness [35].

A comparison of the effects of invasive plants in different regions (in different parts of the secondary range or in regions with different geographic and floristic features) would probably help to better understand the causes of their invasive success. The lack of inter-regional studies of invasive species impacts is highlighted in a major contemporary review [10]. Surely, most field studies of invasive dominants impact assess it in only one part of the secondary range [1–3,5]. At this moment, inter-regional studies of the impact of alien species are mostly performed by comparing such impacts in their native and secondary ranges [36,37]. Inter-regional assessments of the invasion consequences in different parts of the secondary range are rare [35,38].

Here, we compare the taxonomic species richness and proportion of alien plant species in communities invaded by North American ash-leaved maple (*Acer negundo* L.) in different parts of its secondary range. The study was performed in two regions with different geographic and floristic features: the Belarusian Polesia and the Middle Urals. We tested the following three hypotheses:

**Hypothesis 1 (H1).** *In both regions, the species richness (total, herbaceous, and woody) is reduced in A. negundo-invaded communities compared to non-invaded communities dominated by other tree species. This hypothesis has already been tested in both regions, but using different experimental designs. Moreover, only the richness of herbaceous species was assessed in these works [15,38,39].*

**Hypothesis 2 (H2).** *In both regions, the proportion of alien species (total, herbaceous, and woody) does not differ between communities invaded and non-invaded by A. negundo. We make this hypothesis based on the results obtained for invaded communities in the Middle Urals in 2017–2018, when we did not confirm the IMH [40].*

**Hypothesis 3 (H3).** *In the Belarusian Polesia, the proportion of alien species in communities is higher than in communities in the Middle Urals. We make this hypothesis based on the quantifications of the regional alien species pools of the Belarusian Polesia [41] and the Middle Urals [42–49]. These estimates indicate that the regional pool of alien species in the Belarusian Polesia is 1.5 times larger than in the Middle Urals.*

Thus, our first working hypothesis is related to the well-known and often-supported assumption about the overall impact of a strong competitor's invasion on community diversity. Our second hypothesis tested the IMH. Finally, our third hypothesis tested if there is a regional specificity of plant invasions in regions with different climates and alien species pools.

## 2. Materials and Methods

### 2.1. Model Invasive Woody Species—Acer negundo

Ash-leaved maple (*Acer negundo* L.) is a tree that reaches 20–25 m in height and originates in North America. It is distributed from the Rocky Mountains to the Atlantic Coast and from Canada to Florida [50]. This species was introduced to Europe in 1688 and

to Russia in the second half of the 18th century. *A. negundo* is on the list of dangerous invasive species in Europe [51], Belarus [52], Russia [53,54], and the Sverdlovsk Region [55].

In Belarus, ash-leaved maple was first found in the middle of the 19th century. In 1913–1917, it was cultivated in nurseries, including in the surroundings of Gomel. At the moment, *A. negundo* is naturalized throughout Belarus [52] with a high occurrence, especially near human settlements. Moreover, this species forms monodominant flood-plain thickets and an understory of oak, pine, and floodplain forests [52]. At the end of the 19th century, *A. negundo* was introduced to the Middle Urals [56]. At the moment, ash-leaved maple is naturalized in disturbed and semi-natural habitats [53] and urban forest parks [57]. There is an interruption of natural succession and a decrease in species richness in *A. negundo*-invaded communities of the Belarusian Polesia [39]. In the Middle Urals, the dominance of *A. negundo* is followed by a decrease in herbaceous plant species richness [15,40]. However, the IMH was not supported in communities of the Middle Urals [40], and in the Belarusian Polesia, this phenomenon has not been studied.

### 2.2. Study Regions

The first part of the study was performed in the Middle Urals in July 2019 within and around Ekaterinburg (Sverdlovsk Region), with a population of one and a half million (Figure 1, Table 1). The city is located in the south-taiga subzone of taiga. The main vegetation type is coniferous pine (*Pinus sylvestris* L.) forests on sol-podzolic soils and burozems [58]. The climate is cold, with a short warm summer [59] and a six-month-long cold winter with snow cover. About 60%–70% of annual precipitation falls during the warm season (May–August). The modern flora of the Sverdlovsk Region includes 1696 vascular plant species, including 295 alien species. Thus, the alien species proportion in the flora of the Sverdlovsk Region is 17.4% [42–49].

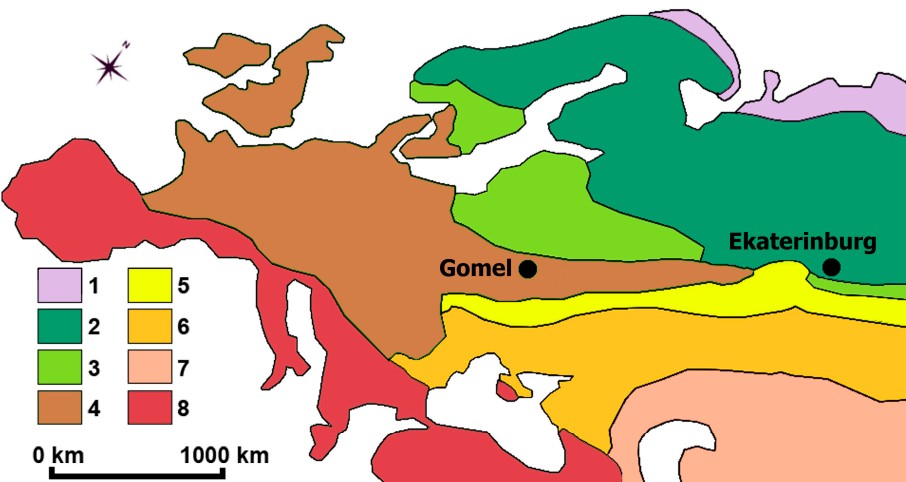

**Figure 1.** Natural zones: 1—tundra and forest tundra; 2—taiga; 3—mixed forests; 4—broad-leaved forests; 5—forest steppes; 6—steppes; 7—semi-deserts and temperate deserts; 8—subtropical hardwood forests.

The second part of the study was performed in the Belarusian Polesia in July 2021 within and around Gomel (Gomelskaya Region), with a population of 502 thousand (Figure 1, Table 1). Modern forest vegetation is coniferous (*P. sylvestris*), mixed, broad-leaved, and deciduous forests. The dominant tree species are *P. sylvestris, Quercus robur* L., *Alnus glutinosa* (L.) Gaertn., *Betula pendula* Roth, *Carpinus betulus* L., *Tilia cordata* Mill., *Acer platanoides* L., and *Fraxinus excelsior* L. The climate is cold, with a warm summer [59]. Maximum precipitation falls in June–July. According to the Information Retrieval System of the Central Botanical Garden of the NAS of Belarus [41], the modern flora of Belarus includes

1964 vascular plant species and 574 alien species. Thus, the alien species proportion in the flora of Belarus is 29.2%.

**Table 1.** Description of the climate, geographical conditions, and floral composition of the regions of the Belarusian Polesia and the Middle Urals.

| Characteristics | Belarusian Polesia, Gomel | Middle Urals, Ekaterinburg |
|---|---|---|
| Coordinates | | |
| Latitude | N 52°20′40″–52°30′20″ | N 56°39′20″–56°57′50″ |
| Longitude | E 30°48.5′30″–31°5′30″ | E 60°22′50″–60°50′30″ |
| Climate [1] | Cold with warm summer (Dfb) | Cold with warm summer (Dfb) |
| Average annual temperature, °C | +4.2 | +3.0 |
| Average temperature in July, °C | +20.4 | +19.0 |
| Average temperature in January, °C | –3.3 | –12.6 |
| Average annual precipitation, mm | 631 | 601 |
| The number of days with temperature above 10 °C | 167 | 132 |
| Soil | Sod-podzolic, sod-podzolic water-logged, peat-bog | Sod-podzolic, burozems |
| Landscape conformation | Plain | Low-mountain |
| Zonal community types | Broad-leaved forests | Coniferous forests |
| Main types of anthropogenic disturbance of vegetation | Deforestation for development and agriculture, fires, recreational use of native forests | |
| Proportion of human-disturbed landscapes, % | 50–60 | |
| Regional plant species richness [2] | | |
| Total | 1964 | 1696 |
| Herbaceous plants | 1758 | 1590 |
| Woody plants | 206 | 106 |
| Native plants | 1390 | 1401 |
| Native herbaceous plants | 1231 | 1312 |
| Native woody plants | 159 | 89 |
| Alien plants | 574 | 295 |
| Alien herbaceous plants | 527 | 278 |
| Alien woody plants | 47 | 17 |

[1] The information about regional climate types is given by [59]. [2] Data about regional plant species richness are given for Belarus [41] and the Sverdlovsk Region [42–49].

### 2.3. Sample Plots

In this work, we used a randomized block design. For vegetation descriptions, we surveyed paired sample plots of 20 × 20 m (block). Block is a site, i.e., an area that includes paired sample plots invaded (An+) and non-invaded (An−) by *A. negundo*. Invaded plots were dominated by *A. negundo*, i.e., the total number of *A. negundo* trunks was the largest compared to any other tree species. In non-invaded plots, the number of *A. negundo* trunks was not the largest compared to the number of trunks of other tree species. Environmental conditions of paired An+ and An− sample plots inside the block were similar, in particular invaded and non-invaded plots: (1) were located in homogenous landscape elements; (2) were close by the degree of anthropogenic transformation; and (3) were located no further than 600 m from each other. One more important criterion of paired sample plot selection was close values of canopy cover. Therefore, the conditions of An+ and An− paired sample plots were similar, except for the dominance of *A. negundo* or other tree species.

In total, we surveyed 12 blocks or 24 paired sample plots in each region. An− sample plots in the Middle Urals were dominated by native *Prunus padus* L., *Pinus sylvestris* L., *Quercus robur* L., *Sorbus aucuparia* L., and *Tilia cordata* Mill., and alien *Malus baccata* (L.) Borkh, *Salix alba* L., *S. fragilis* L., and *Ulmus laevis* Pall. An− sample plots An− in the Belarusian Polesia were dominated by native *Acer platanoides* L., *Alnus glutinosa* (L.) Gaertn., *Fraxinus excelsior* L., *Pinus sylvestris* L., *Populus tremula* L., and *Quercus robur* L., and alien *Robinia pseudoacacia* L.

### 2.4. Vegetation Descriptions

We observed each sample plot by performing vegetation descriptions of vascular plants. We identified the species composition of each species in each layer. Species that did not cause doubts about their taxonomic identification were registered without being collected. Species that were unknown to the authors or caused any doubts about their identification were collected and identified in the lab using dichotomous keys [60,61].

Then, we classified all the species recorded as native or alien in the regions. Alien species are not native to the natural flora of the region, i.e., their introduction was caused by direct or indirect anthropogenic impact. Alien species were identified for each region separately. To classify the species list of the Middle Urals, we used checklists of the Sverdlovsk Region flora [42–49]. The species list of the Belarusian Polesia was classified using the annotated checklist of invasive and potentially invasive species of Belarus [52], the flora checklist of the Biosphere Reserve Pribuzhskoye Polesie [62], and the flora checklist of the Yukhnovsky Biological Reserve [63].

Here, we considered trees and shrubs as a group of woody species. We assessed the parameters of plant communities: (1) species richness (total, woody, and herbaceous)—the number of species recorded in a 400 m² sample plot; (2) native species richness (total, woody, and herbaceous)—the number of native species recorded in a sample plot; (3) alien species richness (total, woody, and herbaceous)—the number of alien species recorded in a sample plot; and (4) proportion of alien species (total, woody, and herbaceous)—the ratio of alien species richness to the total species richness. In the analysis of total and woody alien species richness and proportion, we excluded *A. negundo* from the values of An+ plots.

The species names are given by 'The World Flora Online' [64].

### 2.5. Data Analysis

To designate the regions (the Belarusian Polesia and the Middle Urals) in this section, we used their short names (Polesia and the Urals). To compare species richness parameters, we used the generalized mixed linear model analysis with factors 'plot type' (An+ or An−; *df* = 1), 'region' (Polesia or the Urals; *df* = 1), main factors interaction, and the random factor of block (site with the paired sample plots). In addition, we used the generalized mixed linear model analysis with factors 'plot type', 'region', and 'plant growth form' (woody or herbaceous; *df* = 1) and the random factor of block (site with the paired sample plots) with all possible (double and triple) interactions between the factors. For the analysis, the species richness parameters were logarithmically transformed; the parameters, recorded as proportions, were arcsine transformed. For all the *p*-significance levels, we used FDR-correction; to compute *df*, we used the Kenward–Roger degree of freedom approximation. Significant effects were additionally tested to compare paired differences of means using the Tukey HSD test. When performing analysis, we made sure that our data corresponded to the normality of the distribution (using the Kolmogorov–Smirnov normality test). The graphical visualization was based on the original, non-transformed values. Analysis was performed using Statistica 10, JMP Pro 13, and IBM SPSS Statistics 29.

## 3. Results

### 3.1. Community Composition and Species Occurrence

#### 3.1.1. The Belarusian Polesia

In 24 vegetation descriptions of Polesia, we recorded 254 vascular plant species (55 woody and 199 herbaceous species). The total alien species richness in all 24 descriptions of Polesia was 67 species (26%).

In 12 An− descriptions of Polesia, we recorded 46 woody species at any ontogenetic stage. The most frequent woody species were *A. negundo* (observed in nine descriptions), *Quercus robur* L. (seven descriptions), *Acer platanoides* L., *Fraxinus excelsior* L. (six descriptions), *Crataegus monogyna* Jacq., *Euonymus europaeus* L., *Rubus caesius* L., and *S. aucuparia* (five descriptions). In these descriptions, we observed 167 herbaceous plant species; the most frequent species were *Moehringia trinervia* (L.) Clairv., *Poa palustris* L., *U. dioica* (eight

descriptions), *Taraxacum* sect. *Taraxacum* (seven descriptions), *Calamagrostis epigejos* (L.) Roth, *Carex muricata* L. *Dryopteris carthusiana* (Vill.) H.P.Fuchs, *Galium aparine* L., *Lysimachia vulgaris* L. *Poa trivialis* L., and *Solidago canadensis* L. (six descriptions).

In 12 An+ descriptions of Polesia, we recorded 40 woody species at any ontogenetic stage. The most frequent woody species were *A. negundo* (observed in all 12 descriptions), *A. platanoides* (8 descriptions), *F. excelsior* (7 descriptions), *R. caesius*, *Robinia pseudoacacia* L. (6 descriptions), and *Parthenocissus quinquefolia* (L.) Planch. (5 descriptions). In these descriptions, we observed 120 herbaceous plant species; the most frequent species were *Taraxacum sect. Taraxacum* (observed in all 12 descriptions), *Chelidonium majus* L. (11 descriptions), *Erigeron annuus* (L.) Desf (10 descriptions), *U. dioica* (9 descriptions), *C. muricata*, *G. aparine* (8 descriptions), *Galeopsis* spp. (7 descriptions), *M. trinervia*, *P. palustris*, *Geum urbanum* L., *Artemisia vulgaris* L., *G. hederacea*, *Fallopia convolvulus* (L.) Á.Löve, and *R. repens* (6 descriptions).

3.1.2. The Middle Urals

In 24 descriptions of the Urals, we recorded 192 vascular plant species (43 woody and 149 herbaceous species). The total alien species richness in all 24 descriptions of the Urals was 48 species (25%).

In 12 An− descriptions of the Urals, we recorded 40 woody species at any ontogenetic stage. The most frequent woody species were *A. negundo* (observed in eight descriptions), *Betula pendula* Roth (seven descriptions), *Prunus padus* L., *Malus baccata* (L.) Borkh (six descriptions), *Sorbus aucuparia* L., *Ulmus laevis* Pall., *Amelanchier spicata* (Lam.) K.Koch, *Cotoneaster acutifolius* Turcz., and *Viburnum opulus* L. (five descriptions). In these descriptions, we observed 139 herbaceous plant species; the most frequent species were *G. hederacea, U. dioica* (observed in all 12 descriptions), *Poa angustifolia* L., *Taraxacum* sect. *Taraxacum* (10 descriptions), *Arctium tomentosum* Mill., *Deschampsia cespitosa* (L.) P.Beauv., *Geum aleppicum* Jacq. (9 descriptions), *Ranunculus repens* L. (8 descriptions), *Carum carvi* L., *Stellaria media* (L.) Vill. (7 descriptions), *Agrostis capillaris* L., *Dactylis glomerata* L., *Plantago major* L., *Poa pratensis* L., and *Ranunculus monophyllus* Ovcz. (6 descriptions).

In 12 An+ descriptions of the Urals, we recorded 25 woody species at any ontogenetic stage. The most frequent woody species were *A. negundo* (observed in all 12 descriptions), *P. padus*, *Ribes uva-crispa* L. (5 descriptions), *M. baccata*, *Rubus idaeus* L. (4 descriptions), *B. pendula*, *A. spicata*, and *V. opulus* (3 descriptions). In these descriptions, we observed 85 herbaceous plant species; the most frequent species were *U. dioica*, *Taraxacum* sect. *Taraxacum* (10 descriptions), *G. aleppicum* (9 descriptions), *G. hederacea*, *Catolobus pendulus* (L.) Al-Shehbaz (8 descriptions), *A. tomentosum*, *S. media* (7 descriptions), *A. capillaris*, and *Leonurus quinquelobatus* Gilib. (6 descriptions).

*3.2. Species Richness*

In the text below, ±SE are given for species richness parameter mean values. Community species richness varies significantly between sample plots invaded and non-invaded by *A. negundo* (Figure 2a, Table 2). In An− plots, we observed 40.8 ± 3.9 plant species per 400 m$^2$ in Polesia and 40.3 ± 3.2 in the Urals. In An+ plots, total plant species richness decreased significantly: we observed 32.2 ± 2.5 and 22.9 ± 3.3 species per 400 m$^2$ in Polesia and the Urals, respectively. Therefore, plant species richness in communities invaded by *A. negundo* decreased by 21%–43% compared to the non-invaded plots. Moreover, total species richness varied differently in the invaded and non-invaded plots in different regions. Thus, it significantly decreased in plots An− in the Urals, but was not different between plots An+ and An− in Polesia.

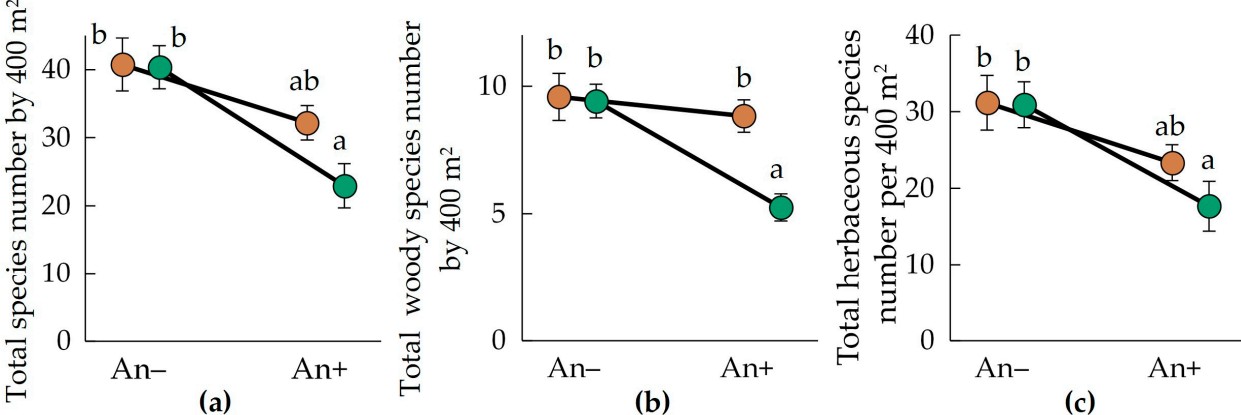

**Figure 2.** Mean (±SE) total (**a**), woody (**b**), and herbaceous (**c**) species numbers in the communities invaded (An+) and non-invaded (An−) by *A. negundo* in descriptions of the Belarusian Polesia (brown circles) and the Middle Urals (green circles). Letters near the circles indicate homogenous groups by the Tukey HSD.

**Table 2.** FDR-adjusted *p*-values of the effects of plot type and region on species richness parameters in the generalized mixed linear model analysis. Significant levels of *p*-values appear in bold.

| Species Number or Proportion | Variability Source | | |
|---|---|---|---|
| | Plot Type (An+ or An−) (*df* = 1) | Region (Urals or Polesia) (*df* = 1) | Plot Type × Region (*df* = 1) |
| Species number per 400 m²: | | | |
| Total | **0.0001** | 0.0714 | **0.0335** |
| Woody | **0.0008** | **0.0063** | **0.0058** |
| Herbaceous | **0.0009** | 0.1438 | 0.0834 |
| Native species number per 400 m²: | | | |
| Total | **0.0001** | 0.4096 | 0.1507 |
| Woody | **0.0001** | **0.0001** | **0.0198** |
| Herbaceous | **0.0002** | 0.7175 | 0.2574 |
| Alien species number per 400 m²: | | | |
| Total | 0.5236 | 0.1777 | **0.0438** |
| Woody | 0.9069 | 0.4948 | **0.0295** |
| Herbaceous | 0.3988 | **0.0118** | 0.2958 |
| Alien species proportion, %: | | | |
| Total | **0.0335** | 0.5718 | 0.2610 |
| Woody | 0.5929 | 0.1800 | 0.1890 |
| Herbaceous | **0.0349** | **0.0158** | 0.3073 |

Similarly, we observed decreased woody (by 8%–44% in the invaded plots; Figure 2b, Table 2) and herbaceous species richness (by 25%–43% in the invaded plots; Figure 2c, Table 2).

Woody species richness was greater in the region of Polesia. Moreover, woody species richness varied differently in the invaded and non-invaded plots in different regions. Thus, it significantly decreased in plots An− in the Urals, but was not different between plots An+ and An− in Polesia.

### 3.3. Native Species Richness

Total native species richness varied significantly between the invaded and non-invaded plots, but not between the regions (Figure 3a, Table 2). In An− plots, the native species number was 29.5 ± 3.1 in Polesia and 31.5 ± 3.1 in the Urals. In invaded communities (An+),

native species number decreased by 34%–48%: we observed 19.4 ± 2.1 native species in the invaded plots of Polesia and 16.7 ± 2.8 native species in the invaded plots of the Urals.

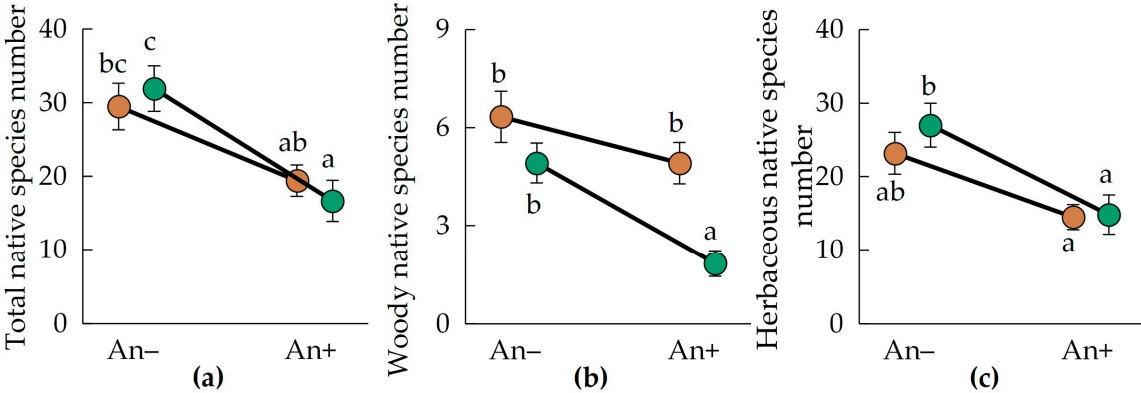

**Figure 3.** Mean (±SE) total (**a**), woody (**b**), and herbaceous (**c**) native species number in the communities invaded (An+) and non-invaded (An−) by *A. negundo* in descriptions of the Belarusian Polesia (brown circles) and the Middle Urals (green circles). Letters near the circles indicate homogenous groups by the Tukey HSD.

In *A. negundo*-invaded communities, observed native woody species decreased by 22%–63% (Figure 3b, Table 2) as did the native herbal richness, which decreased by 37%–45% (Figure 3c, Table 2). Moreover, native woody richness was significantly greater in the region of Polesia.

### 3.4. Alien Species Richness and Proportion

Total alien species richness varied significantly between invaded and non-invaded plots (Figure 4a, Table 2). In An− plots, the alien species number was 9.0 ± 1.6 in Polesia and 8.3 ± 1.2 in the Urals. In invaded communities (An+), alien species numbers decreased by 24% in the invaded plots in the Urals (6.3 ± 0.8 alien species) and increased by 20% in the invaded plots in Polesia (11.3 ± 1.2 alien species).

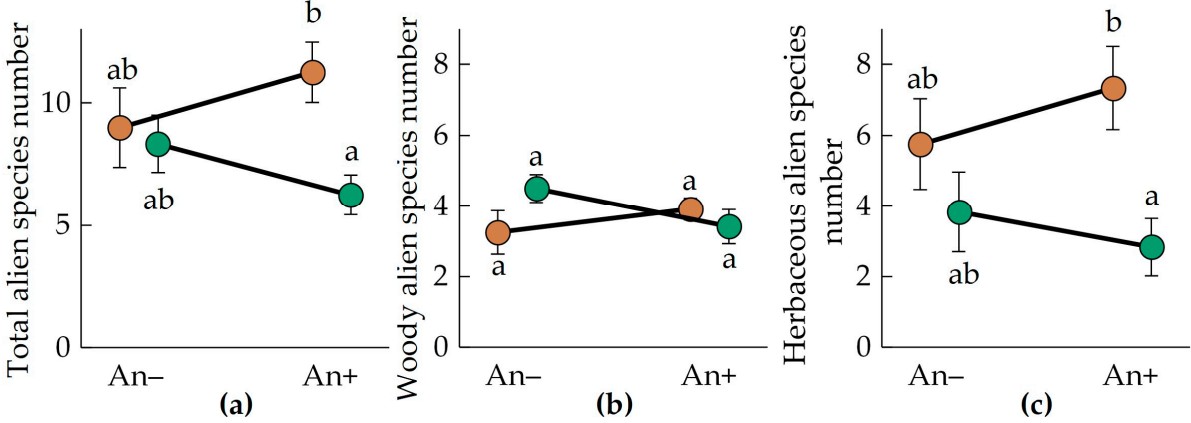

**Figure 4.** Mean (±SE) total (**a**), woody (**b**), and herbaceous (**c**) alien species numbers in the communities invaded (An+) and non-invaded (An−) by *A. negundo* in descriptions of the Belarusian Polesia (brown circles) and the Middle Urals (green circles). Letters near the circles indicate homogenous groups by the Tukey HSD.

Woody alien species richness varied significantly between invaded and non-invaded plots in different regions (Table 2). Alien woody species numbers varied between 3.4 and 4.5 species, respectively, but the difference between the compared groups was not significant (Figure 4b).

Average alien herbaceous species richness was relatively greater in Polesia (6.5 ± 0.9 species per 400 m$^2$) compared to the Urals (3.3 ± 0.7 species per 400 m$^2$) (Figure 4c, Table 2).

In 48 vegetation descriptions, average alien species proportions were close to 25%. The average alien species proportion of the total species number was greater in invaded plots (Table 2). In An− plots, the alien species proportion was 22% ± 4% in Polesia and 21% ± 3% in the Urals. In invaded communities (An+), the alien species proportion was 33% ± 3% in Polesia and 26 ± 4 in the Urals. But the difference between the compared groups was not significant (Figure 5a). The proportion of alien woody species did not depend on the analyzed factors (Figure 5b, Table 2).

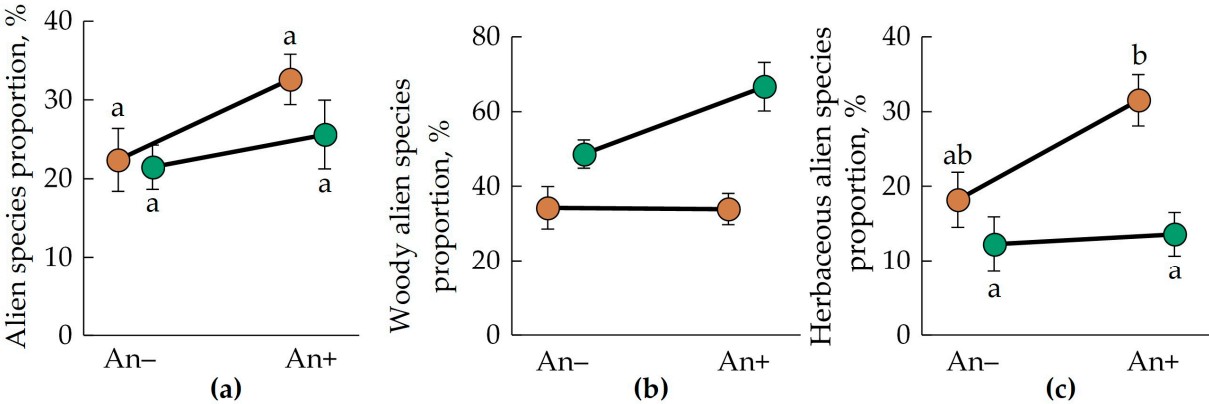

**Figure 5.** Mean (±SE) total (**a**), woody (**b**), and herbaceous (**c**) alien species proportion in the communities invaded (An+) and non-invaded (An−) by *A. negundo* in descriptions of the Belarusian Polesia (brown circles) and the Middle Urals (green circles). Letters near the circles indicate homogenous groups by the Tukey HSD.

The proportion of alien herbaceous species varied significantly between the regions and was twice as much in Polesia (25% ± 3%) than in the Urals (13% ± 2%). Significant differences were found between the plots invaded and non-invaded by *A. negundo*, too: An− plots had 15% ± 3% of alien herbaceous species. In An+ plots, the alien herbaceous species proportion was greater—23% ± 3%.

Moreover, we found an interesting pattern. Alien woody species proportion was greater in the Urals (45% ± 4%) than in Polesia (35% ± 3%). At the same time, the proportion of alien herbaceous species was greater in Polesia (25% ± 3%) than in the Urals (13% ± 2%). This pattern is significant. We performed the generalized mixed linear model to assess the effects of plot type (*df* = 1), region (*df* = 1), and plant growth form (woody or herbaceous, *df* = 1) on alien species percentage. In this analysis, we found significant effects for plant growth type ($F_{(1;75)} = 30.52$; $p < 0.0001$) and the interaction of the effects of the region and plant growth form ($F_{(1;75)} = 10.90$; $p = 0.0015$).

## 4. Discussion

In general, our results show that:

- The plant species number in the communities invaded by *A. negundo* in both study regions decreased compared to the non-invaded communities;
- Apparently, this is primarily related to a decrease in the number of native woody and herbaceous species in the *A. negundo*-invaded communities;
- The alien species proportion is greater in the Belarusian Polesia than in the Middle Urals.

### 4.1. Acer Negundo Invasion-Induced Species Richness Decrease

Our first working hypothesis, that community species richness decreased as a result of *A. negundo* invasion, is confirmed. In both study regions, we found that ash-leaved maple



invasion is accompanied by a decrease in community species richness. This result, firstly, was partly expected, and secondly, we can find its consistent explanation in a significant number of other papers on plant invasion effects. The novelty of this result is that we have found, in general, the same effects of *A. negundo* invasion on plant species richness for different growth forms (woody and herbaceous) in two different regions of Eurasia.

Information about the negative consequences of invasions on native species diversity (including plant species richness) is numerous [1–5]. Such results are known for *A. negundo* [15,39,40]. There are fewer works describing the opposite result when plant invasions did not decrease or even increase plant species richness in recipient communities [6]. The plant richness decrease was found both in paired block experimental designs, i.e., in works comparing invaded and non-invaded communities [65], and in designs assessing community responses to invasion throughout the invader abundance gradient [5,66]. In our case, the effect of *A. negundo* invasion was detected not only when comparing the dominated by *A. negundo* (invaded in our study) communities and the communities without its domination (non-invaded in our study). We additionally performed the Generalized Linear Model (GLM) analysis, where we used the factor 'proportion of *A. negundo* trunks' (*df* = 1) instead of the factor 'plot type' to assess the effect of *A. negundo* invasion throughout its abundance gradient (see Supplementary File S1: Table S1; Figures S1 and S2). We found that most of the results were, in general, similar, or at least consistent, in both analysis methods, including the results of the effects of *A. negundo* on total and native species richness.

Mechanisms of negative effects of *A. negundo* on native communities are likely related to its ability to be a transformer species. These features of ash-leaved maple are studied in the Middle Urals and relate to its ability to:

- Form high shading [15,16] due to its high canopy cover [15];
- Increase soil moisture [67];
- Inhibit the development of arbuscular mycorrhiza [68].

Probable allelopathic effects of *A. negundo* on the decrease in plant species richness are less likely [69].

### 4.2. Native-Alien Species Richness Relationship

In our second working hypothesis, we assumed that the proportions of native and alien species in the communities invaded or non-invaded by *A. negundo* were similar. This hypothesis was based on the results we obtained in the Middle Urals, where the IMH in the communities invaded by *A. negundo* was not confirmed [40]. Testing this hypothesis with our new data, we obtained interesting but not absolutely clear results. We found that the total and herbaceous alien species proportions in the communities invaded by ash-leaved maple increased. Moreover, in general, the decrease in total species richness in both study regions comes from a decrease in native species richness. That is, the response to *A. negundo* invasion effects, expressed as a decrease in species richness, is common to native species but not to alien woody and herbaceous species. Similar effects were detected both in the paired analysis and in the analysis of the *A. negundo* abundance gradient (see Supplementary File S1: Table S1; Figures S3 and S4). This result appears to be the first, though not exhaustive, confirmation of the invasional meltdown hypothesis for *A. negundo*. Therefore, our second hypothesis was not confirmed. We believe it is an unexpected and interesting result.

Unfortunately, it is difficult to hypothesize some real mechanisms, such as the selective effects of ash-leaved maple on native and alien plant species. The IMH suggests the presence of such mechanisms [19–21]. To explain them, it is necessary to have additional data about the biological and ecological features of a large number of native and alien species. However, such data for the studied regions is unknown. But it is important to highlight that one of the detailed properties of alien species in the Belarusian Polesia is their ability to form arbuscular mycorrhiza [70]. A study of 19 alien and 25 native plant species found that alien plant arbuscular mycorrhiza formation is less effective than native plant arbuscular mycorrhiza formation [70]. Probably, using the hypothesis about one of

the *A. negundo* invasion consequences, suppression of mycorrhiza [68], we can explain the high alien herb proportion by the fact that alien species are less dependent on mycorrhyzal symbiosis than the native ones. So, alien species in the communities invaded by *A. negundo* seem to have a competitive advantage over native species. Surely, this hypothesis needs a special justification. Additionally, it is important to analyze the biological and ecological functional traits of alien species, such as life cycles, dispersal strategies, shade tolerance, etc., to understand the probable effects assumed by the IMH.

### 4.3. Levels of Invasion in Different Regions

In our third working hypothesis, we assumed a difference in the levels of invasion between communities in two investigated regions. This hypothesis was based on the species pool hypothesis, which states that there is a positive relationship between the richness of the local community and the richness of the regional species pool [29–32]. In this study, we hypothesized that levels of invasion in the communities of the Belarusian Polesia are greater than in the communities of the Middle Urals. Our hypothesis was based on the data about the richness of regional species pools in these regions. Thus, the regional species pool of alien herbaceous plants in the Belarusian Polesia is nearly two times greater than in the Middle Urals. The alien woody species pool in Belarusian Polesia is almost three times greater than in the Middle Urals.

We found that alien herbs in the communities of the Belarusian Polesia have almost two times greater species richness than alien herbs in the communities of the Middle Urals. However, alien tree species richness was greater in the communities of the Middle Urals. Thus, our hypothesis about the relationship between community species richness and regional species pool richness was confirmed for herbaceous plants but not for woody plants. Apparently, the species pool hypothesis cannot be considered fully supported by our data. This result is only partially explained and requires further study.

### 5. Conclusions

In general, we obtained the following three main results:

1. In the regions with different geographic and floristic conditions (the Belarusian Polesia and the Middle Urals), plant species richness in the communities invaded by *Acer negundo* decreased compared to the non-invaded communities. The differences in species richness between the invaded and non-invaded communities in different regions are not equal but have the same directions. This result shows that the negative impact of *A. negundo* on community species richness is similar on an inter-regional scale, i.e., in different parts of its secondary range.

2. In different parts of *A. negundo* secondary range, its invasion primarily caused a decrease in native plant species richness. This phenomenon may be interpreted as evidence of the presence of weak effects predicted by the invasional meltdown hypothesis in invaded communities. We believe that this result is new in relation to the invasion of *A. negundo*. This result indicates that *A. negundo* may have selective impacts on native and alien plants.

3. Average herbaceous plant invasion levels in the communities of the Belarusian Polesia are greater than in the communities of the Middle Urals. But the average levels of invasion of alien woody plants in the communities of the Belarusian Polesia are lower than in the communities of the Middle Urals. This result highlights the importance of climatic and geographical conditions, land use history, and disturbance regimes in different regions for the distribution and sustainability of alien species in local communities.

**Supplementary Materials:** The following supporting information can be downloaded at: https://www.mdpi.com/article/10.3390/f14112118/s1, Table S1: FDR-adjusted p-values of the effects of *Acer negundo* trunk proportion and region on species richness parameters in Generalized Linear Model analysis; Figure S1: Relationship between species numbers and proportions of trunks of *A. negundo* in descriptions of Belarusian Polesia and the Middle Urals; Figure S2: Relationship between native species numbers and proportions of trunks of *A. negundo* in descriptions of Belarusian Polesia and the Middle Urals; Figure S3: Relationship between alien species numbers and proportions of trunks of *A. negundo* in descriptions of Belarusian Polesia and the Middle Urals; Figure S4: Relationship between alien species proportions and proportions of trunks of *A. negundo* in descriptions of Belarusian Polesia and the Middle Urals.

**Author Contributions:** Conceptualization, D.V.V.; methodology, D.I.D. and D.V.V.; formal analysis, D.I.D. and D.V.V.; investigation, D.I.D.; data curation, D.I.D.; writing—original draft preparation, D.I.D. and D.V.V.; writing—review and editing, D.I.D., D.V.V. and A.P.G.; visualization, D.I.D.; supervision, D.V.V. All authors have read and agreed to the published version of the manuscript.

**Funding:** This research was funded by the Russian Science Foundation, grant number 23-24-00645 (https://rscf.ru/project/23-24-00645/ (accessed on 6 October 2023)). The APC was funded by the Russian Science Foundation grant number 23-24-00645 (https://rscf.ru/project/23-24-00645/ (accessed on 6 October 2023)).

**Data Availability Statement:** Terms of access to the original data used in the study are discussed with the corresponding author upon request.

**Acknowledgments:** The authors are grateful to N.V. Zolotareva (Institute of Plant and Animal Ecology of the Ural Branch of the Russian Academy of Sciences) for the help in identification of the herbarized material; Dubrovina D.P. (Institute of Plant and Animal Ecology of the Ural Branch of the Russian Academy of Sciences) for the help in carrying out the field phase of research in Ekaterinburg; Sokolov A.S. (Francisk Skorina Gomel State University) for helping in the selection of test areas in and around Gomel; and Sozontov A.N. (Institute of Plant and Animal Ecology of the Ural Branch of the Russian Academy of Sciences) for the discussion about the used methods of data analysis. The authors are also grateful to the two anonymous reviewers who pointed out the errors and inaccuracies in the text and the contents of the manuscript and helped to significantly improve it during the review.

**Conflicts of Interest:** The authors declare no conflict of interest.

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
