# Peer review of "Plant Species Richness and Invasional Meltdown in Different Parts of Acer negundo L. Secondary Range"

_forests, doi:10.3390/f14112118_

Round 1

Reviewer 1 Report

The manuscript presents some new and quite interesting information on the ecological impact of the invasive tree species Acer negundo on native vegetation at two locations. Such research is definitely needed to assess the ecological impact of invasive species and maybe find strategies to minimize it. Relatively few studies are done comparing two geographically very distant areas with the same methodology, and this is one of the strong points in the presented manuscript. Although the two areas (Belarus and Middle Ural) share a number of ecological parameters (such as total species richness climatic conditions), they differ in the richness of alien herbal and woody plants, which is a crucial factor in this analysis. The use of paired sample plots to compare invaded and uninvaded areas which otherwise share the same vegetation type makes sense to highlight the impact of this one single crucial parameter. However, I believe that more information on the plots, especially in respect to the presence of A. negundo, is needed and should be considered for deeper analysis and discussion. Invaded plots were dominated by A. negundo, whereas in uninvaded plots the number of A. negundo trunks was not the largest…(lines 140 ff). That means, that the An- plots were not always free of A. negundo. Actually, 9 of 12 An- plots in Polesia had A. negundo (line197-198), and 8 of 12 An- plots in Middle Ural (lines 219-220). I think it is critical to know the exact abundance of A. negundo in the different plots to better understand its impact on native and alien vegetation, because the difference between most An+ and An- plots seems not to be a simple presence/absence of the crucial factor A. negundo, but a more complicated and varied relative presence of that factor. The data could be presented in a supplemental file. I also think it would be interesting to compare the ‘real’ An- plots (the plots without any A. negundo; 3 in Polesia and 4 in the Middle Urals) with the An+ plots and the plots with ‘minor presence’ of A. negundo. I think that would significantly increase the scientific value of the manuscript and maybe also help to get a better idea about some open questions raised in the discussion, such as the importance of IMH (lines 343 ff).

I suggest asking a native English speaker for critical reading and editing the manuscript. The manuscript contains many grammar mistakes which have to be corrected. E.g.: lines 92ff: It (is) distributed from (the) Rocky Mountains to (the) Atlantic Coast.... This species (was) introduced (to) Europe... 

Reviewer 2 Report

The manuscript submitted by the authors appears to be a good piece of scientific work: the introduction and discussion are supported by relevant and up-to-date studies, the methodology is of high quality, and the statistical data analysis (considering interactions and data transformations) is commendable. However, I have two major reservations about the manuscript:

Firstly, concerning the data analysis, in my opinion, it could have been conducted using more sophisticated methods. Secondly, although of lesser importance, the language editing of the text may need improvement. Occasionally, it is evident that the text was not written by a native speaker. I have noticed some more noticeable errors, and I point them out specifically. However, as I am not a native speaker myself, there might be more, and I recommend having the text reviewed by a native speaker.

L138: If you have used a block design in which pairs of samples were created (commendably), you could have employed generalized linear mixed models for your analysis, where such a pair would define a random term.

L180: It is commendable that a two-way ANOVA, including the testing of interactions, was utilized. However, I still have reservations about this. There are two fundamental requirements for conducting an analysis of variance (ANOVA): Homoscedasticity (homogeneity of variance): This requirement demands that the variances in different groups (levels of the independent variable) are approximately equal. Normality of data: Another requirement is that the data follow a normal distribution (Gaussian distribution). If there is a substantial amount of data (> 60), you can deviate from this assumption if you can assume at least normality of the model residuals.

I have serious concerns that the dependent variables may not meet this normality of residuals requirement because they are integer numbers. In cases where you have discrete data, normality of residuals mostly cannot be achieved, ANOVA assumptions may thus be violated, and authors might consider using alternative statistical methods, such as logarithmic transformation or (better) the aforementioned generalized linear models (with Poisson or negative binomial distribution). For proportions, beta regression, rather than arcsine transformation, may be more appropriate in that case.

L183: "all possible interactions" … Just double, or also triple interactions (between all factors)?

L190-L191: I think that in the manuscript, much earlier (e.g., in the methodology), you could have mentioned how you would abbreviate both locations. Not including this information in a one-sentence paragraph at the beginning of the results, where it seems cumbersome to me.

Figure 6: We can observe much larger variances for Belarusian Polesia compared to Middle Urals – naturally, in the second case, we have only zeros. Thus, both ANOVA assumptions are likely violated in this case – homoscedasticity and data normality. Therefore, a different method of data analysis should be considered at least in this scenario.

L61-L63: "In this moment, inter-regional studies of alien species impact mostly performed by comparing such impact in their native and secondary range [36–37]." → "AT this moment, inter-regional studies of the impact of alien species ARE mostly performed by comparing such impacts in their native and secondary ranges."

L94: "A. negundo is in the list of dangerous invasive species in Europe" → “on the list” is correct

L105: "was not support" → “was not supported”

L142-L143: "In uninvaded plots the number of A. negundo trunks was not the largest in compare with the number of other tree species trunks. " → "In uninvaded plots, the number of A. negundo trunks was not the largest when COMPARED/IN COMPARISON to the number of trunks from other tree species."

L159-L160: "Registered without collecting" → "were registered without collecting"

L187: "Analysis performed" → "Analysis was performed"

L197, L206, L219, and L230: The first sentences of the paragraphs are repetitively structured similarly (which seems fine to me) with minor deviations that are grammatical errors – the correct variant is on L230, while there appears to be a verb missing in some sentences (L197, L219), or there are incorrect word orders (L206) – please revise (a unified structure should be sufficient).

L239-L240: "Community species richness vary significantly between invaded and uninvaded by A. negundo sample plots" → "Community species richness varies significantly between sample plots invaded and uninvaded by A. negundo."

L256-L257: "…but was not differ between plots An+ and An– in Polesia." → "was not differing/different"?

L265-L266: "as well as native herbal richness – decreased by 37–45%" → "as did the native herbal richness, which decreased by 37–45%"

L277: The phrase "varied between 3.4 and 4.5" suggests a range (for one phenomenon), so adding "respectively" is redundant and could be confusing, or vice versa, „respectively“ is correct here, but „varied between 3.4 and 4.5 makes no sense, in that case.

L312-L313: "…is decreased in compare with uninvaded communities" → "in comparison"

L316: "he" → "The"

L319: "that community species richness is decrease…" → "decreased"?

L393-L394: "is decrease in compare with" → "has decreased compared to."

Round 2

Reviewer 1 Report

The revised version of the manuscript is significantly improved to the original one and deserves publication.

Reviewer 2 Report

The manuscript has undergone English language revision and the incorporation of most of the comments from both reviewers. Thanks to this, it has been significantly improved. However, I cannot resist adding two more comments regarding the statistical analysis. 

Authors claim: "We could indeed use generalized linear mixed models with random term of a pair, but in this analysis the variability related to the characteristics of different regions would also be random, what is obviously wrong: inter-regional variability is not random and caused by a large number of factors related to the geographical, floristic and socio-economic characteristics of regions. Therefore, in analyzing the data we deliberately abandoned the use of generalized mixed linear models with random factor and used two-way ANOVA." My response to this comment: It is evident that interregional variability is influenced by a multitude of factors. The authors' explanation appears perplexing, especially considering that this is precisely why blocks or a randomized block design (as mentioned by the authors themselves) are commonly employed as random terms. When dealing with blocks (paired samples) sourced from the same location while there exists substantial variability across different locations, incorporating a random term for blocks allows for the consideration of block effects, mitigating the potential variability introduced by location-specific factors. Hence, if significant variability exists between different locations and the authors aim to account for this variability in their analysis, it is logical to utilize a random term for blocks within a (G)LMM rather than opting for a regular analysis of variance (ANOVA). This approach enables the model to recognize that paired samples were collected within the same blocks and that specific factors may influence the outcomes. Such an approach can substantially enhance the validity of the statistical analysis.

Authors claim: "For the sample of the proportion of alien tree species, the Levéne’s test demonstrated an inequality between the variances of the compared groups. But in this analysis we did not found any significant differences, i.e. this analysis has no decisive influence on our conclusions. So, we did not exclude this analysis and did not change its method." My response: If the authors adhere to using ANOVA instead of methods that would account for the design they devised themselves, they should at least consider using the Kruskal-Wallis test instead of ANOVA in situations where the assumptions of ANOVA are not met. Retroactively justifying the use of ANOVA in situations where the criteria for ANOVA are not met by claiming that it did not yield significant results anyway demonstrates a misunderstanding of the philosophy of statistical analysis. If the assumptions of ANOVA, such as normality of residuals and homoscedasticity, are not met, it is usually inappropriate to present the results of ANOVA, even with the argument that the results were not significant anyway.
